# Study of the Structure and Mechanical Properties after Electrical Discharge Machining with Composite Electrode Tools

**DOI:** 10.3390/ma15041566

**Published:** 2022-02-19

**Authors:** Timur Rizovich Ablyaz, Evgeny Sergeevich Shlykov, Karim Ravilevich Muratov, Ilya Vladimirovich Osinnikov

**Affiliations:** Department of Mechanical Engineering, Perm National Research Polytechnic University, 614000 Perm, Russia; kruspert@mail.ru (E.S.S.); karimur_80@mail.ru (K.R.M.); ilyuhaosinnikov@bk.ru (I.V.O.)

**Keywords:** electrical discharge machining, microhardness, roughness, residual stresses, mechanical properties, chemical composition

## Abstract

Our study was devoted to increasing the efficiency of electrical discharge machining of high-quality parts with a composite electrode tool. We analyzed the chemical composition of the surface layer of the processed product, microhardness, the parameter of roughness of the treated surface, residual stresses, and mechanical properties under tension and durability with low-cycle fatigue of steel 15. Our objective was to study the effect of the process of copy-piercing electrical discharge machining on the performance of parts using composite electrode tools. The experiments were carried out on a copy-piercing electrical discharge machining machine Smart CNC using annular and rectangular electrodes; electrode tool materials included copper, graphite, and composite material of the copper–graphite system with a graphite content of 20%. The elemental composition of the surface layer of steel 15 after electrical discharge machining was determined. Measurements of microhardness (HV) and surface roughness were made. Residual stresses were determined using the method of X-ray diffractometry. Metallographic analysis was performed for the presence of microdefects. Tensile tests and low-cycle fatigue tests were carried out. The mechanical properties of steel 15 before and after electrical discharge machining under low-cycle fatigue were determined. We established that the use of a composite electrode tool for electrical discharge machining of steel 15 does not have negative consequences.

## 1. Introduction

One of the prioritized areas of mechanical engineering is the manufacture of high-quality products. Their operating conditions are constantly getting tougher. There is a need to use spatially complex structures in the design. To improve the reliability of products, modern materials with high mechanical and physical properties are used. The use of these materials makes it possible to increase the operational characteristics of the manufactured products.

Despite the advantages of using materials with increased mechanical and physical properties in mechanical engineering, significant wear of the cutting tools occur during their traditional blade processing. Additionally, when cutting along the path of a complex profile, it becomes necessary to purchase additional equipment. One example of a rational electrophysical method of processing is the method of electrical discharge machining (EDM) [1].

EDM consists of changing the shape, size, roughness, and properties of the surface of the workpiece under the influence of electrical discharges as a result of electrical erosion [2]. To carry out the EDM process, it is necessary to create a high concentration of energy in the discharge zone. A pulse generator is used to achieve this goal. The current pulses generated by the pulse generator are applied to the work electrode and the electrode tool (ET). The EDM process takes place in a working fluid, a dielectric, which fills the interelectrode space [3]. When the electric field strength in a certain zone of the interelectrode gap exceeds the critical value, a breakdown of the interelectrode gap will occur. The breakdown of the interelectrode space is a plasma channel that quickly heats up to ultra-high temperatures (Figure 1). As a result, the material’s microvolume melts and evaporates. Particles of molten material are thrown into the interelectrode gap and solidify in the form of sludge particles, which are washed out by the flow of a dielectric liquid [4].

The allowance removed from the workpiece during EDM is formed as a result of the superposition of single erosion holes. The post-EDM surface is formed by a set of overlapping wells.

During the EDM process, significant changes occur in the surface layer of the workpiece. The post-EDM surface layer can be conditionally divided into several zones (Figure 2): the zone of saturation with elements of the working fluid, the zone of deposition of ET material, the white layer formed from molten ET material, the heat affected zone, and the zone of plastic deformation. The sequence of the formation of zones and their number, structure, and properties largely depend on the material being processed, the processing mode used by the working fluid, the ET material, and the conditions of the process. As a rule, there is no clear difference between the zones, and in most cases, they overlap each other [5,6].

The EDM method is widely used in the processing of materials with enhanced physical and mechanical properties. It allows the derivation of complex-shaped products from conductive materials of any hardness. However, this method is not without its drawbacks [7,8,9]. EDM is characterized by low performance and high ET wear. This increases the cost of the resulting products. An adequate solution to this problem is the use of ETs with increased electrical discharge properties. At the moment, a number of composite materials have been developed that can significantly increase the operational properties of ETs. According to [10,11], an electrode made of a mixture of copper and colloidal graphite has the best balance of performance and wear resistance. However, its influence on the microstructure of the surface layer and the mechanical properties of the processed product remains unexplored.

An urgent scientific and technical problem is the experimental analysis of the effect of EDM using composite ETs on the mechanical properties and structure of the surface layer of the workpiece.

### Related Work

Currently, there is a lot of research in the field of EDM. The main directions of research surrounding the EDM process that were observed in our literary analysis are shown in Table 1.

Based on our analysis of the literature, we concluded that the dynamics of research have changed (Figure 3). There has been an increase in the amount of research on EDM in general over the past 15 years. The largest number of studies is devoted to changes in surface morphology and topography during EDM, and the rate of their development is greatest. Over the past 5 years, there has been a sharp increase in the number of studies that focus on changing the chemical composition of the treated surface. There is much less research on the mechanical properties and the resulting white layer after EDM. Nevertheless, their number has also increased several times.

Leading universities around the world are engaged in EDM research. The chemical composition and structure of the processed surface in comparison with traditional copper and graphite electrodes remains unexplored.

## 2. Materials and Methods

### 2.1. Materials and Methods

ETs for EDM were manufactured (Figure 4 and Figure 5). The ETs were made in the form of a ring for the study of mechanical properties after EDM. To conduct the experiment, an ET with dimensions of 20 × 20 × 5 mm was made from a composite material based on copper and a preparation of dry colloidal graphite (PNRPU, Perm, Russia). For the manufacture of an ET blank from a composite material, the method of powder metallurgy was used. After receiving, the workpiece was processed to the required dimensions by milling. Composite ET are made on the basis of copper powders and dry colloidal graphite preparation. For the manufacture of a workpiece from a composite material, the powder metallurgy method was used: copper powder was mixed with a preparation of dry colloidal graphite. The effectiveness and relevance of the application of this ET is explained in [34,35,36]. In [37], the performance, wear resistance, and accuracy of ETs from various composite pseudo alloys based on copper were studied: Cu-Cr (copper-chromium); Cu-Mo (copper-molybdenum); Cu-W (copper-tungsten); Cu-C (copper-colloidal graphite), etc., depending on the percentage of components in the EDM alloy steel. It was shown that of all the studied materials, the best balance of electroerosive properties (productivity, accuracy, wear resistance) was possessed by ETs from a composite material of the copper-colloidal graphite system with a graphite content of 20%.

For tensile and low-cycle fatigue tests, ring-shaped electrode tools were used (Figure 5).

Structural carbon steel 15 was used to study the structure, chemical composition, and mechanical properties of the treated surface.

The processed samples were a flat body of the plate type with a thickness of 5 mm. On the samples, grooves were made using the copy-piercing EDM method with various ETs on a copy-piercing EDM Smart CNC at finishing and rough processing modes (Table 2). The processing depth was 3 mm. The EDM processing of tensile and low-cycle fatigue test specimens was performed in an orbital cycle.

### 2.2. Study of the Machined Surface

To study the change in the elemental composition of the surface layer of steel 15 in the process of EDM, the method of X-ray spectral analysis was used. The measurements were carried out on an REM-100U scanning electron microscope (Electron, Sumi, Russia).

Microsections were made to study the microstructure. Sections were made in two stages. At the first stage, the samples were pressed into the Top Tech Presidon (Top tech machines Co., LTD., Taichung, Taiwan) automatic assembly press. At the second stage, sanding was carried out on emery paper with grit sizes from p240 to p1500 on a Top Tech Plato grinding machine (Top tech machines Co., LTD., Taichung, Taiwan). To reveal the structure, the microsection was etched with a 4% solution of nitric acid in ethyl alcohol.

The microstructure on the microsections and treated surface were examined using an OLYMPUS GX 51 light microscope (Olympus corporation, Tokyo, Japan) at magnifications up to 1000. Image processing was performed using the OLYMPUS Stream Motion software (Olympus corporation, Tokyo, Japan).

Microdurometric tests were carried out on a PMT-3 microhardness meter (Lomo, Saint Petersburg, Russia) in accordance with the requirements. The hardness on a PMT-3 microhardness tester was determined by the method of the restored indentation by indentation of a four-sided diamond pyramid with a square base. The applied load was 50 g. The shutter speed was 6 s. The measurements were carried out in accordance with the requirements. The calculated value of microhardness is translated by the formula:(1)$$HV=\frac{0.102\times 2F\times sin\frac{\alpha }{2}}{d^{2}}=0.189\frac{F}{d^{2}}$$
where *F* is the force, N.
(2)$$HV=\frac{2P\times sin\frac{\alpha }{2}}{d^{2}}=1.854\frac{F}{d^{2}}$$
where *P* is the weight, kgs; α is the angle between opposite faces at the vertex, equal to 136°; and *d* is the arithmetic mean of the lengths of both diagonals of the imprint; after removing the load, mm.

The roughness of the processed surface was measured using a Mahr Perthometer S2 profilometer (Carl Mahr Holding GmbH, Esslingen, Germany) in accordance. The base length was 0.8 mm.

The following parameters were measured: average roughness height (*Ra*), maximum roughness height (Rmax), and average roughness step (Sm).

### 2.3. Measurement of Residual Stresses in the Surface Layer

The measurements were carried out in accordance with the technique described in [38]. Determination of the magnitude of residual stresses of the 1st kind according to the classification of N.N. Davidenkov (RS_1_) was carried out by the X-ray diffractometry method using the Xstress 3000 robotic complex (Stresstech Oy, Jyväskylä, Finland) considering the parameters of the material given in Table 3.

Modes of OH measurement by X-ray diffractometry are presented in Table 4.

Mathematical processing of the RS measurement results was carried out in the Xtronic diffractometer control program. Mathematical processing parameters are given in Table 5.

Measurements were made on EDM surfaces as well as untreated areas.

In our study, measurements were taken on the outer surface of the sample. The RS level was determined at three points for each machined groove.

### 2.4. Tensile Testing and Low-Cycle Fatigue

Tensile and low-cycle fatigue test methods were developed in accordance. To determine the main physical and mechanical characteristics of the material, solid samples with a circular cross section were used. The fixation of the samples in the testing machine was carried out using hydraulic wedge grips. In Figure 7, photographs of the appearance and a sketch of the samples subjected to static and cyclic tests are presented.

For these tests, four samples were made by the methods of EDM and turning. This allowed for the comparison of the effects of the processing method on the mechanical properties of the resulting product. Three samples were fabricated with EDM using copper, graphite, and composite ETs to study the effect of the ET material on the properties of the machined part. EDM was carried out in draft mode.

All experiments were carried out at normal temperature on an Instron 8850 biaxial servohydraulic test system (Norwood, MA, USA) focused on dynamic and static tests, a general view of which is shown in Figure 8. The testing machine is equipped with a Dynacell two-axis electronic load cell with a load range of ±160 kN for axial loading, ±1 kN/m for torsion, and a measuring accuracy of 0.4%. A dynamic axial displacement transducer with a strain measurement range of ±40% and an accuracy of 0.5% was used to measure deformations during tensile testing.

Cyclic tests were carried out without a strain gauge with stress control (stress range—σP = 500 MPa, asymmetry coefficient—Rσ = 0) and a given frequency of 0.5 Hz. Low-cycle fatigue experiments were carried out until the samples broke into two parts.

## 3. Results

### 3.1. Elemental Composition of the Processed Material

The results of studying the elemental composition of samples before and after EDM with various ETs in rough and finishing modes are presented in Table 6. Based on the data obtained, histograms were built (Figure 9).

Analysis of the diagrams showed that during the EDM process, the manganese content in the surface layer decreases, regardless of the ET material. Moreover, at a weaker finishing mode, these changes are more pronounced. In EDM with a graphite ET, silicon is completely removed from the surface layer. When processing with a copper ET in the finishing mode, its content does not change. In draft mode, it is halved.

Particular attention should be paid to the copper content in the surface layer. In the case of a copper ET, the amount of copper particles increased significantly. This was due to the saturation of the surface layer of the part with ET material. By dielectric flows, molten particles of ET material enter the melting zone of the workpiece material and mix with it.

In the case of a composite electrode, also containing copper, the change in its content during the EDM process was less significant. In the finishing mode, saturation of the surface layer with copper was not observed. In the rough mode, during processing, it was transferred to the workpiece material. However, its content was almost three times less than after treatment with a copper ET. No increase in graphite content was observed.

Thus, the use of composite ETs does not increase the variation in the surface layer chemistry of the workpiece material. In contrast, there is less transfer of copper from the ET to the melting zone of the workpiece material.

### 3.2. Microstructure and Surface of Samples after EDM

The results of examining the surface of steel 15 for the presence of microcracks after EDM in the finishing mode are presented in Figure 10.

The results of examining the surface of steel 15 for the presence of microcracks after EDM in the rough mode are presented in Figure 11.

The analysis of the results obtained showed that the surfaces obtained by EDM using an ET from a copper-colloidal graphite composite material are closely similar to the surfaces obtained by processing ETs from copper; there is no fundamental difference in the number of microcracks.

The results of studying the surface structure of steel 15 after EDM in finishing mode are presented in Figure 12.

The results of the study of the surface structure of steel 15 after EDM in the rough mode are presented in Figure 13.

It was found that in all cases, regardless of the ET material, the regularity of the formation of the thickness and structure of the white layer was preserved. In the finishing mode of processing, a uniform white layer with a thickness of 1–3 microns was formed along the entire length of the surface. In the rough mode of processing, the white layer is intermittent (there is a large number of material breaks); the thickness of the white layer itself can vary from 0 to 6 microns.

This phenomenon can be explained by the fact that the physical essence of the EDM process, the change in the structure of the treated surface, and the formation of a white layer depend on the energy processes in the breakdown channel. From a physical point of view, the transfer of discharge energy to electrodes is determined by the movements of particles and molecules, which can be divided into two types: the movement of charged particles under the action of an external electric field and thermal movement. The flare component is of the greatest importance for the formation of the structure and the presence of a white layer. Under the action of the discharge, the surface of the ET instantly heats up to the boiling point and above, which leads to the ejection of torch vapors at speeds that are much higher than the speed of sound. Reaching the opposite electrode, the torch jet transfers thermal energy to the surface. In the presence of a temperature difference between the two electrodes, the material of one of the electrodes is transferred, in a vapor state, to the surface of the other electrode.

### 3.3. Microhardness of the Surface Layer

Based on the obtained data on the microhardness of the surface layer of the material being processed, diagrams of microhardness according to Vickers (HV) were plotted depending on the depth of measurement.

The results of the study of the microhardness of the surface of steel 15 after EDM in the finishing mode are presented in Figure 14.

The results of the study of the microhardness of the surface of steel 15 after EDM in the rough mode are presented in Figure 15.

During the EDM of steel 15 in the zone of the white layer at a depth of up to 150 microns, a 25–35% increase in the surface microhardness was observed. At a depth of 150 to 1000 microns, the microhardness decreased by 25–35%, which was associated with tempering from thermal influence. At a depth of more than 1000 microns, the hardness of the material is stabilized.

It has been established that a drop in the level of microhardness can be seen on the surface of the samples. This phenomenon may be caused by the heating of the surface of the part during EDM. This occurs as a result of additional tempering under noticeable heating during the action of the ET. The minimum value of the thermal observation zone was observed when processing in the strong range. This can be explained by the short time of exposure to the surface of the electrolyte part due to the high productivity of processing.

No significant influence of ET material on the process of microhardness alteration during EDM has been established.

### 3.4. Surface Roughness

Surface roughness parameters processed by EDM copper graphite and composite ET are presented in Table 7.

Surface profiles after EDM in finishing mode are shown in Figure 16.

Surface profiles after EDM in draft mode are shown in Figure 17.

Based on the analysis of profilograms, histograms were constructed (Figure 18).

It was found that regardless of the ET material, the average (Ra) and maximum microroughness heights (Rmax) and the average roughness pitch were within the same range. The difference between the maximum and minimum indicators was less than 20%.

The presence of the roughness parameter in EDM from a physical point of view can be explained by the uneven transfer of energy from the spark discharge to the ET material of the part. It is possible to observe an uneven distribution of the electric field in the breakdown channel and, as a consequence, an uneven generation of thermal energy. The energy that is on ETs is total and consists of several components. These are electronic, ionic, torch, gas-kinetic, radiant, and volumetric. The torch, electronic, and ionic components have the greatest influence on the formation of the value of the roughness parameter. Flare transfer of the processed material occurs in the vapor state and is caused by the temperature difference between the ET and the workpiece. The uneven formation of the total energy components occurs during the EDM of materials with different thermophysical properties. These properties directly affect the electrical erosion resistance of these materials and, as a result, the formation of a macrorelief. Thus, the finding of the roughness parameters in the same range can be explained.

It is shown that the use of a composite material of the copper-graphite system as an ET material leads to a critical decrease in the roughness parameters.

### 3.5. Residual Stresses

The results of measuring the residual stresses before and after EDM with copper, graphite, and composite ETs are presented in Figure 19.

In the diagrams, the change in the sign of the residual stresses after the EDM can be seem, regardless of the direction of measurement and the material of the ET. This is due to a change in the direction of residual stresses in the surface layer of the material being processed. It is shown that residual stresses in the EDM process acquire a tensile character. This, in addition to some other factors such as surface pits, can cause cracking.

ET material has a negligible effect on the residual stress value. The residual stresses generated by composite ET EDM are comparable to the residual stresses generated by EDM using ETs made from traditional materials.

### 3.6. Tensile Properties

The results of our study of the effect of turning and EDM using various ETs on the mechanical properties of steel 15 are presented in Table 8.

Based on the results obtained, a histogram of the effect of the treatment method on the mechanical properties in tension was constructed (Figure 20).

Based on the results obtained, a histogram of the effect of the processing method on the mechanical properties under tension was constructed.

Analysis of the histogram showed a slight change in the mechanical properties of the samples machined on a lathe and EDM, regardless of the ET material. The difference in the results obtained does not exceed 10%. Figure 21 shows photographs of tensile fracture surfaces of processed specimens.

It was shown that the destruction of experimental specimens under tension occurred mainly in the center of the workpiece, regardless of the technology of their manufacture (Figure 22).

Figure 21 shows the view of the destroyed samples obtained by different operations. It can be seen that after EDM, there is a characteristic surface on the surface of the workpiece, characterized by a large number of single holes superimposed on each other. However, it should be noted that the overall pattern of destruction is the same. The destruction of the samples took place under the influence of a cyclic load.

It is worth noting the conventional yield stress. The performance of sample processed by EDM using a composite ET exceeds that of one using a copper or graphite ET. This indicator determines at what ultimate loads the product material passes from an elastic state to a plastic one. The higher this indicator, the larger chance that the product will be able to return to its original state after removing the load. Accordingly, the likelihood of occurrence and development of cracks in the surface layer and the base material as a whole is reduced.

### 3.7. Durability of Samples under Low-Cycle Fatigue

Table 9 presents the results of our study of the durability of the clouds obtained by different methods for low-cycle fatigue.

Based on the analysis of the data obtained, it was found that the durability of steel 15 samples obtained by the EDM method with low-cycle fatigue was 30% less in comparison to the samples obtained on a lathe. The sample treated with the composite electrode showed low-cycle fatigue life close to the EDM average.

Figure 23 shows photographs of the sample fracture surface under cyclic loading after turning and EDM.

The areas of destruction of specimens during turning and EDM found during testing for low-cycle fatigue are the same and are located in the areas of the meeting of straight and radius surfaces. Thus, it is not the ET material, but the physics of the EDM process itself that has the greatest impact on the fracture of EDM samples.

## 4. Conclusions

Possible negative uses of an ET from a composite material such as a pseudo-alloy of the copper and colloidal graphite system at EDM of steel 15 were investigated.The influence of composite ETs on the chemical composition, microstructure, and mechanical properties of steel 15 in comparison with traditional copper and graphite ETs was studied. Comparison of samples was processed by traditional blade methods.The analysis of the treated surface of the EDM showed an uneven distribution of the heat-affected zone. An uneven distribution of microhardness in the depth of processing was observed, depending on the processing modes. A decrease in the value of microhardness near the treated surface was noted. This phenomenon is explained by excessive overheating of the treated surface. It should be noted that at maximum processing conditions, microhardness decreased. This phenomenon may be associated with the peculiarity of the material being processed, as well as with the intensive productivity of the processing process, as a result of which the modified surface layer was removed from the processing zone.The formation of the roughness of the treated surface after EDM is formed by superimposing single holes onto each other. The size and shape of a single well depends on the energy of the pulse, as well as on the material of the instrument electrode and the material being processed. The unevenness of the formation of the roughness of the machined surface depends on the uneven distribution of the pulse energy. It has been established that the torch component of a single discharge has the greatest influence on the process of roughness formation. Under the action of high temperatures in the breakdown channel, heating, evaporation, and torch transfer of the processed material occur.

The uneven distribution of energy depends on many factors. One of the main factors affecting the nonuniformity is the difference in the thermophysical properties of the materials of the ET and the part.

5.It was found that the use of composite ETs does not have negative consequences. On the contrary, in a number of parameters, composite ETs are superior to traditional copper and graphite. The possibility of using composite ETs for EDM steel 15 was demonstrated.

## Figures and Tables

**Figure 1 materials-15-01566-f001:**
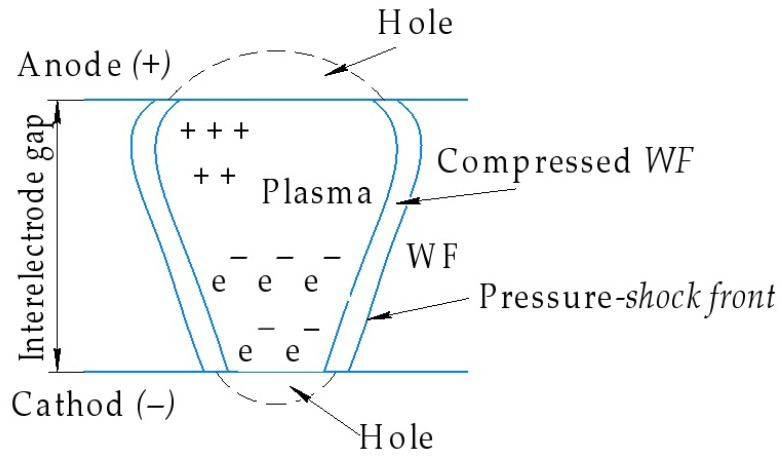
Plasma channel.

**Figure 2 materials-15-01566-f002:**
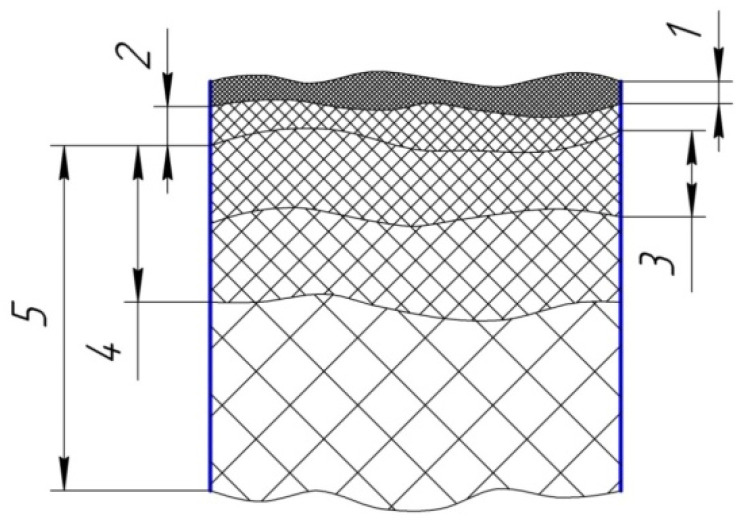
Schematic location of the zones of the surface layer after the EDM: 1—zone of saturation with elements of the working fluid; 2—zone of deposition of the material of the electrode-tool; 3—white layer formed from molten workpiece material; 4—heat affected zone; 5—zone of plastic deformation.

**Figure 3 materials-15-01566-f003:**
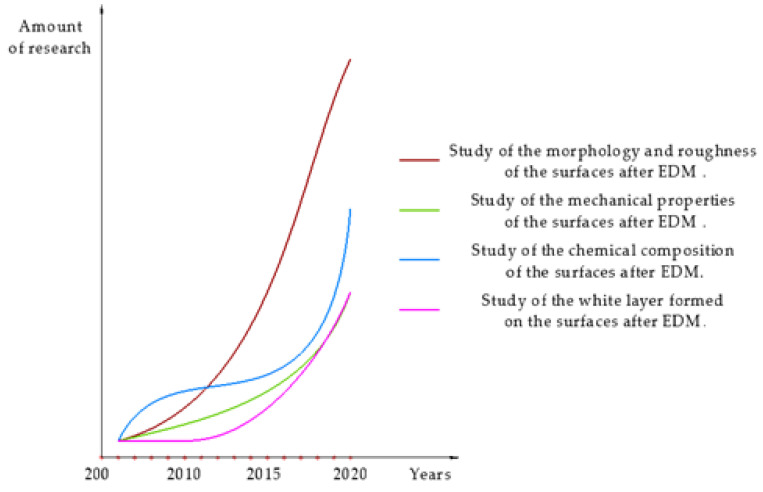
Dynamics of research in the field of EDM.

**Figure 4 materials-15-01566-f004:**
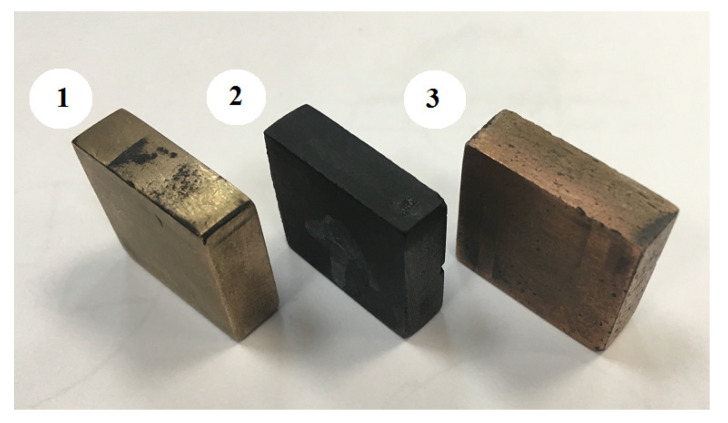
Electrode tools: 1—copper, 2—graphite, 3—composite.

**Figure 5 materials-15-01566-f005:**
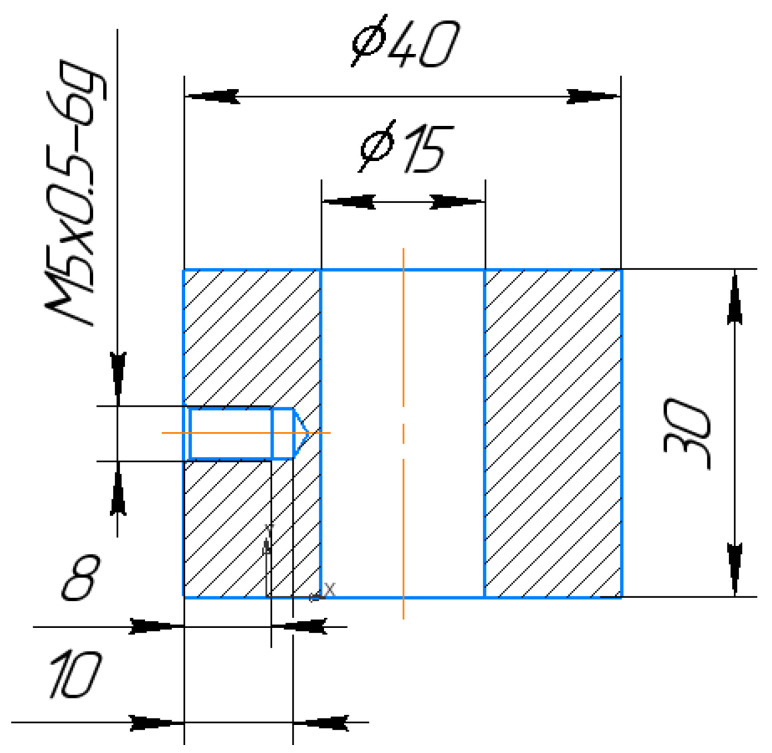
Ring-shaped electrode tools.

**Figure 6 materials-15-01566-f006:**
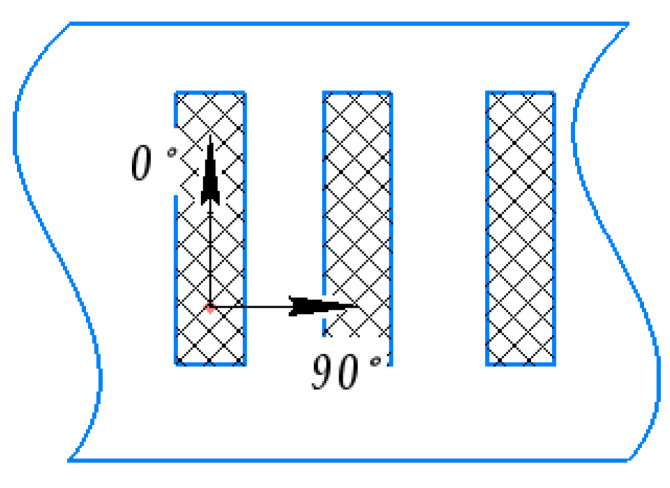
Direction diagram φ to the point of analysis.

**Figure 7 materials-15-01566-f007:**
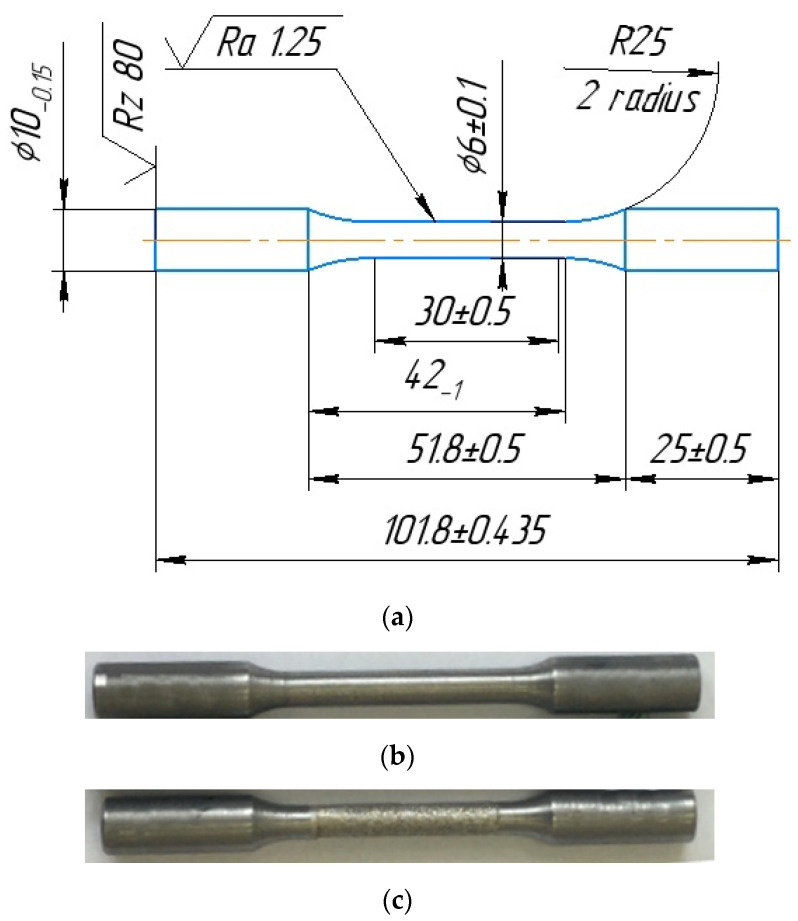
Sketch (**a**) and photographs of samples, the working parts of which were processed by turning (**b**) and EDM (**c**) methods.

**Figure 8 materials-15-01566-f008:**
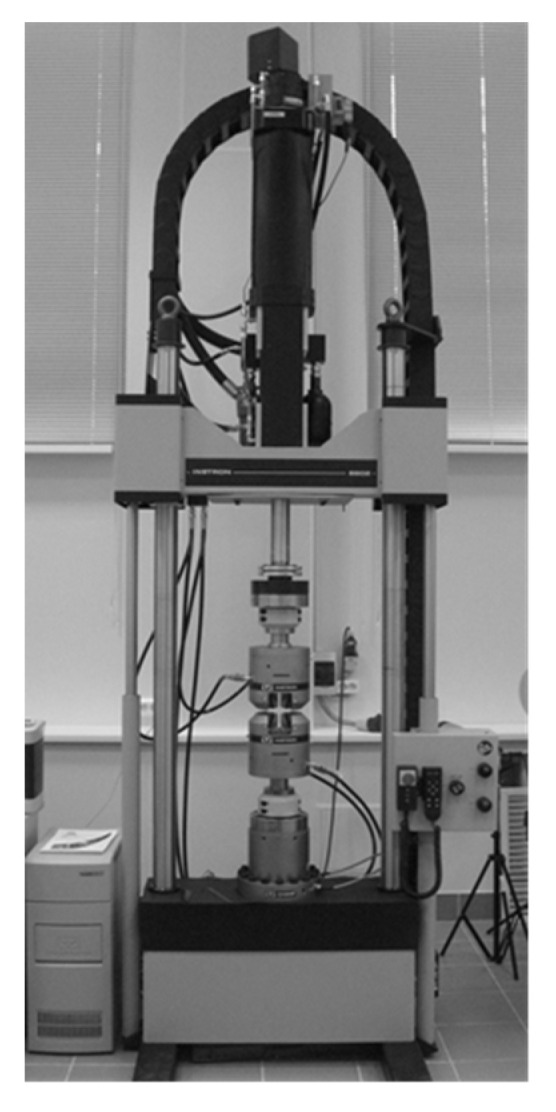
Instron 8850 servohydraulic test system.

**Figure 9 materials-15-01566-f009:**
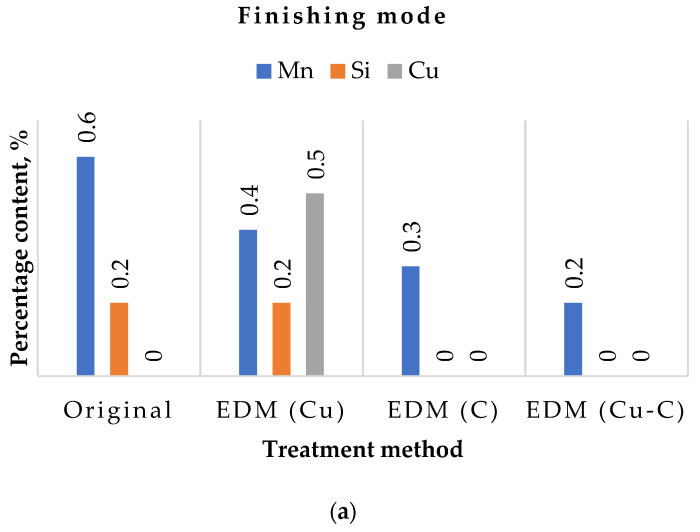
Elemental composition of samples before and after EDM: (**a**) in finishing mode; (**b**) in draft mode.

**Figure 10 materials-15-01566-f010:**
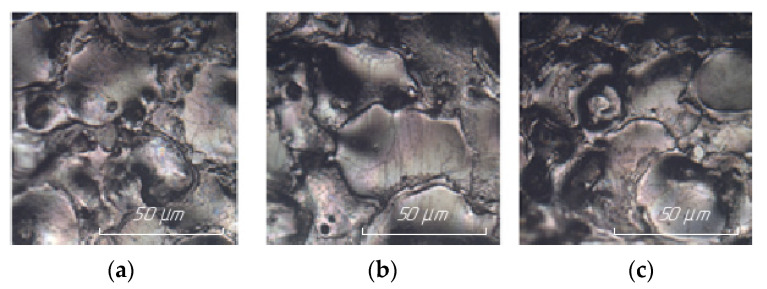
Surface of steel 15 at 200× magnification after EDM in ET finishing mode from: (**a**) copper; (**b**) graphite; (**c**) composite.

**Figure 11 materials-15-01566-f011:**
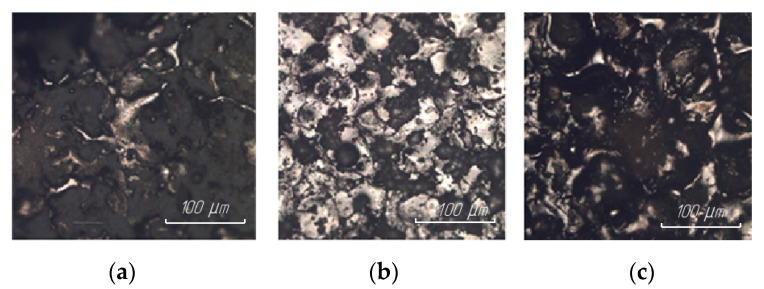
Surface of steel 15 at 100× magnification after EDM in rough ET mode from: (**a**) copper; (**b**) graphite; (**c**) composite.

**Figure 12 materials-15-01566-f012:**
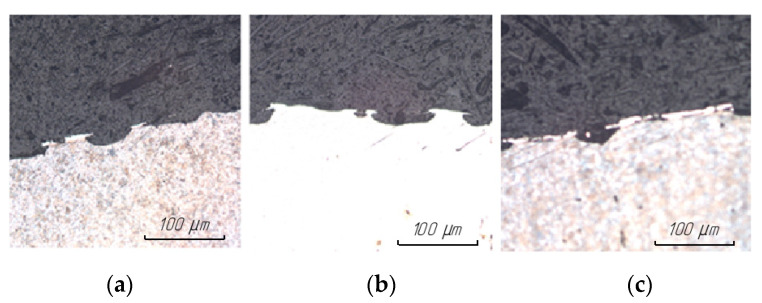
Surface structure of steel 15 at 100× magnification after EDM in ET finishing mode from: (**a**) copper; (**b**) graphite; (**c**) composite.

**Figure 13 materials-15-01566-f013:**
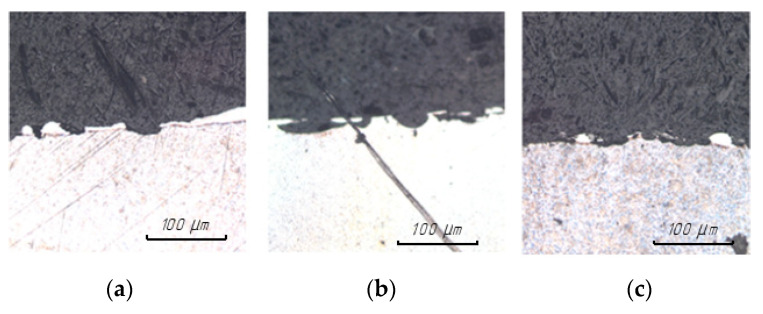
Surface structure of steel 15 at 100× magnification after EDM in rough ET mode from: (**a**) copper; (**b**) graphite; (**c**) composite.

**Figure 14 materials-15-01566-f014:**
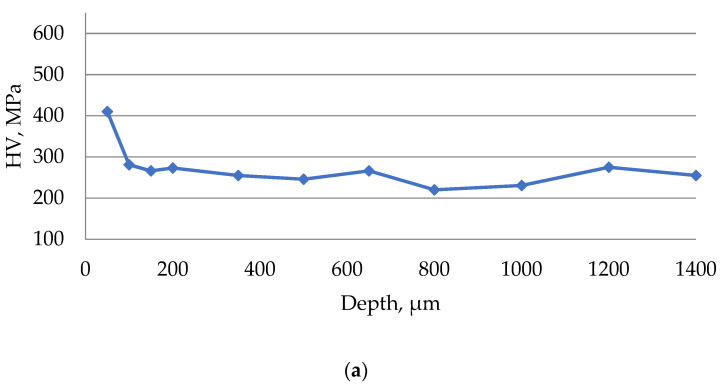
The results of measuring the microhardness of the surface of steel 15 after EDM in the finishing mode using ETs from: (**a**) copper; (**b**) graphite; (**c**) composite.

**Figure 15 materials-15-01566-f015:**
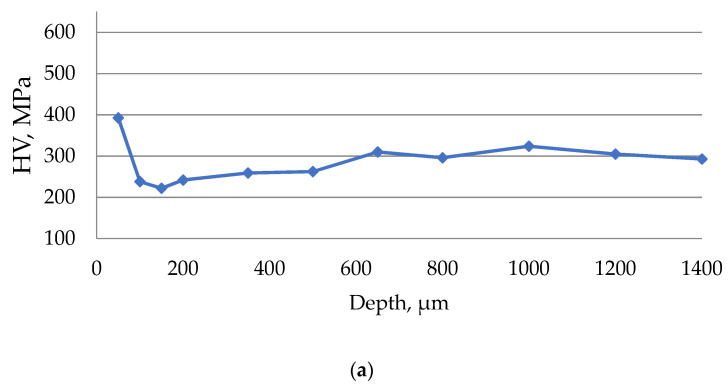
The results of measuring the microhardness of the surface of steel 15 after EDM in the rough mode using ETs from: (**a**) copper; (**b**) graphite; (**c**) composite.

**Figure 16 materials-15-01566-f016:**
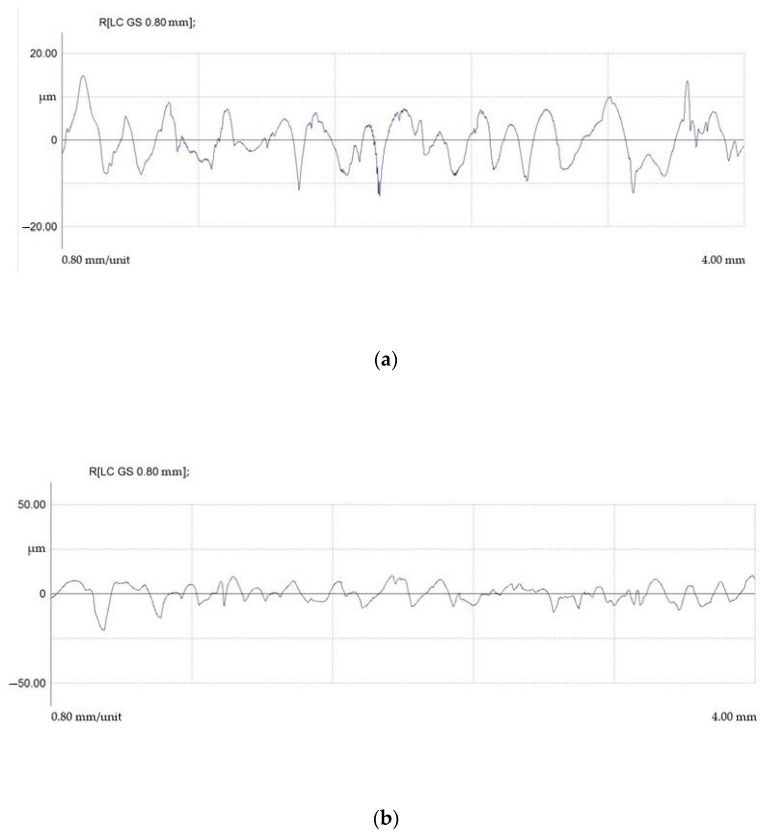
Profilograms of the surface of steel 15 after EDM in finishing mode using ETs from: (**a**) copper; (**b**) graphite; (**c**) composite.

**Figure 17 materials-15-01566-f017:**
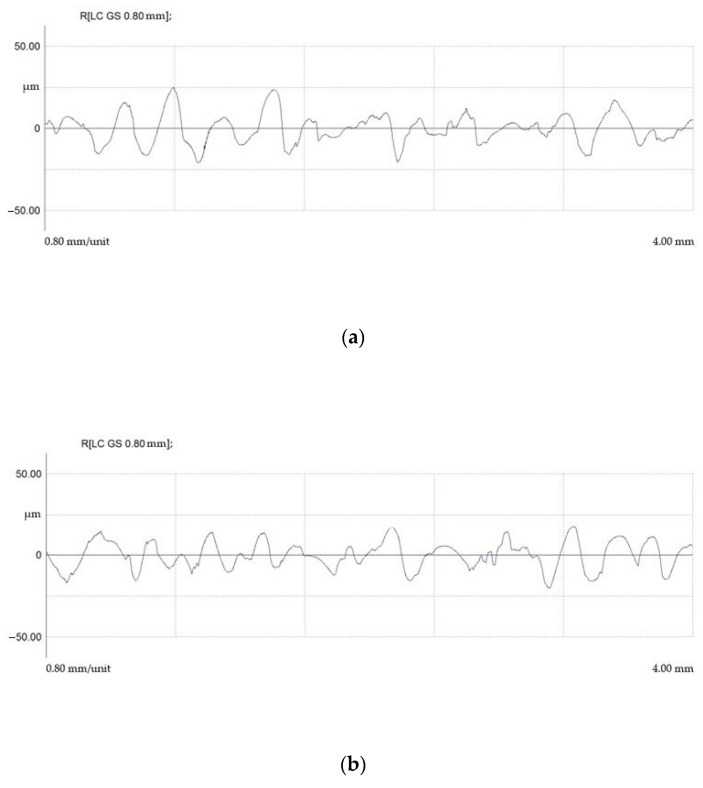
Profilograms of the surface of steel 15 after EDM in rough mode using ETs from: (**a**) copper; (**b**) graphite; (**c**) composite.

**Figure 18 materials-15-01566-f018:**
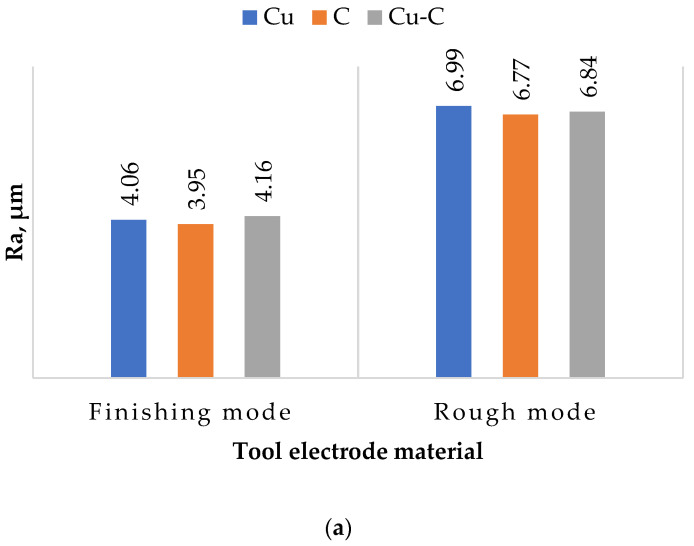
Histograms of surface roughness parameters after EDM with different ETs: (**a**) average roughness height (Ra); (**b**) maximum roughness height (Rmax); (**c**) average roughness pitch (Sm).

**Figure 19 materials-15-01566-f019:**
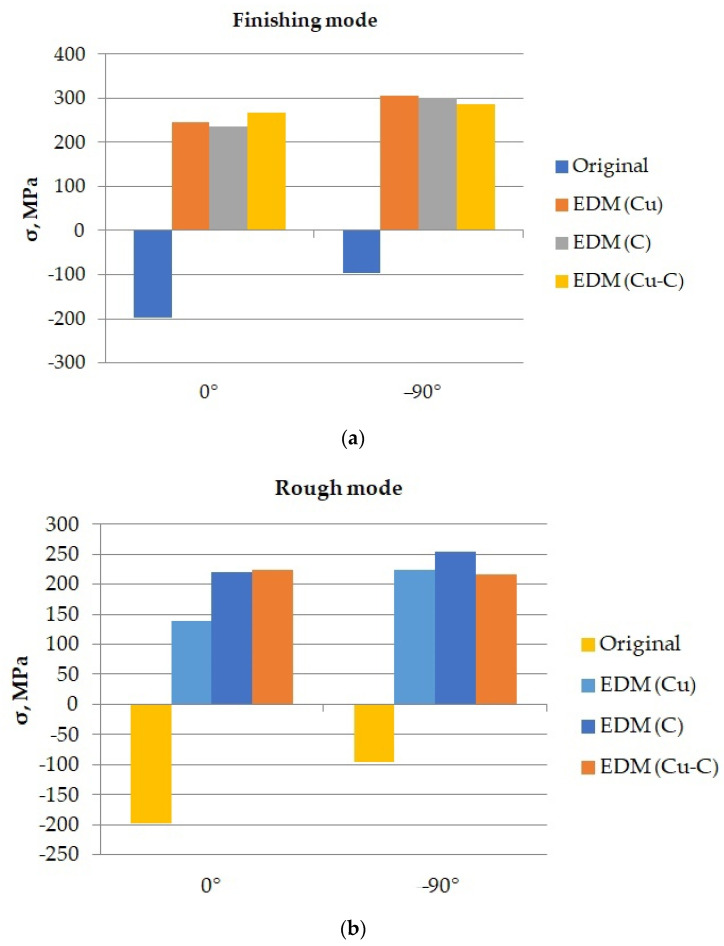
Histograms of changes in the residual stresses of the samples in the EDM process: (**a**) in the finishing mode; (**b**) in draft mode.

**Figure 20 materials-15-01566-f020:**
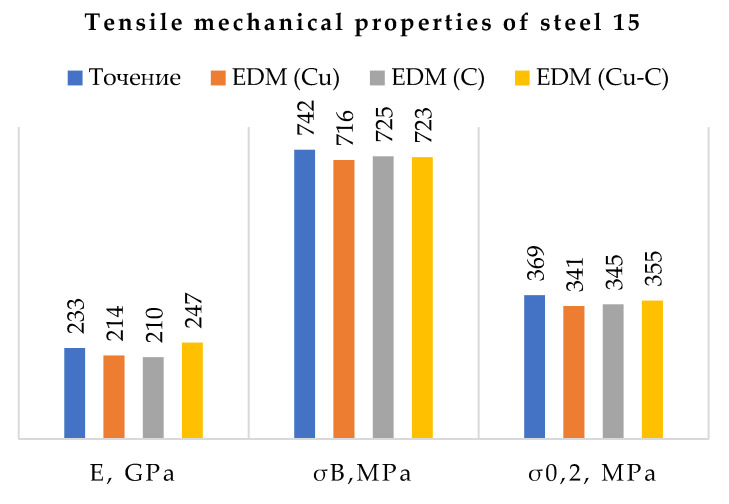
Bar graph of the change in the mechanical properties of steel 15 under tension.

**Figure 21 materials-15-01566-f021:**
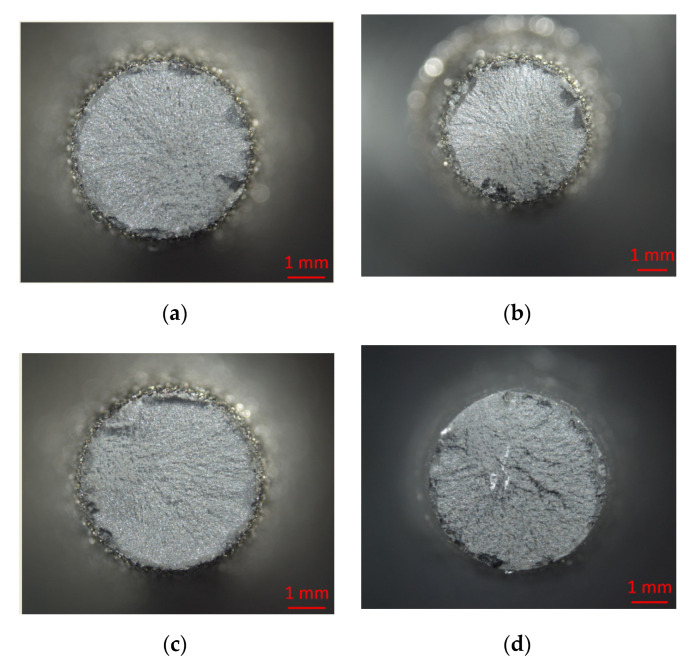
Photographs of the fracture surface of tensile specimens manufactured by: (**a**) EDM (copper ET); (**b**) EDM (composite ET); (**c**) EDM (graphite ET); and (**d**) turning.

**Figure 22 materials-15-01566-f022:**
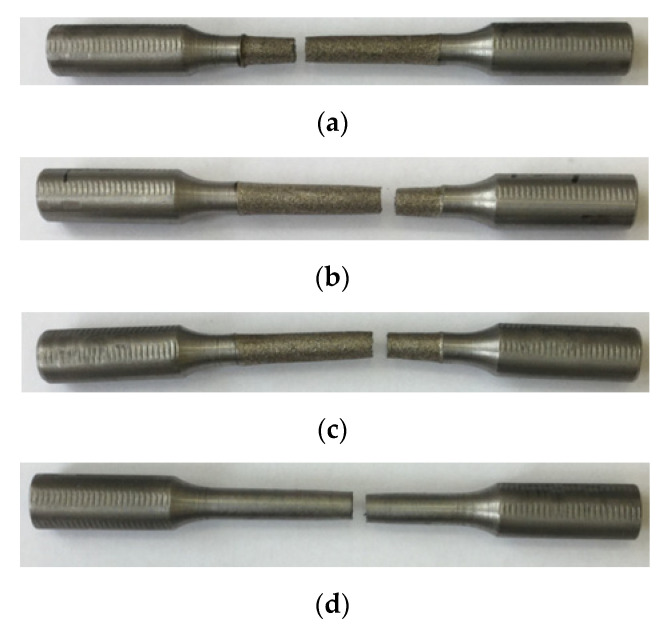
Photos of tensile fractured specimens made by: (**a**) EDM (copper ET); (**b**) EDM (composite ET); (**c**) EDM (graphite ET); (**d**) turning.

**Figure 23 materials-15-01566-f023:**
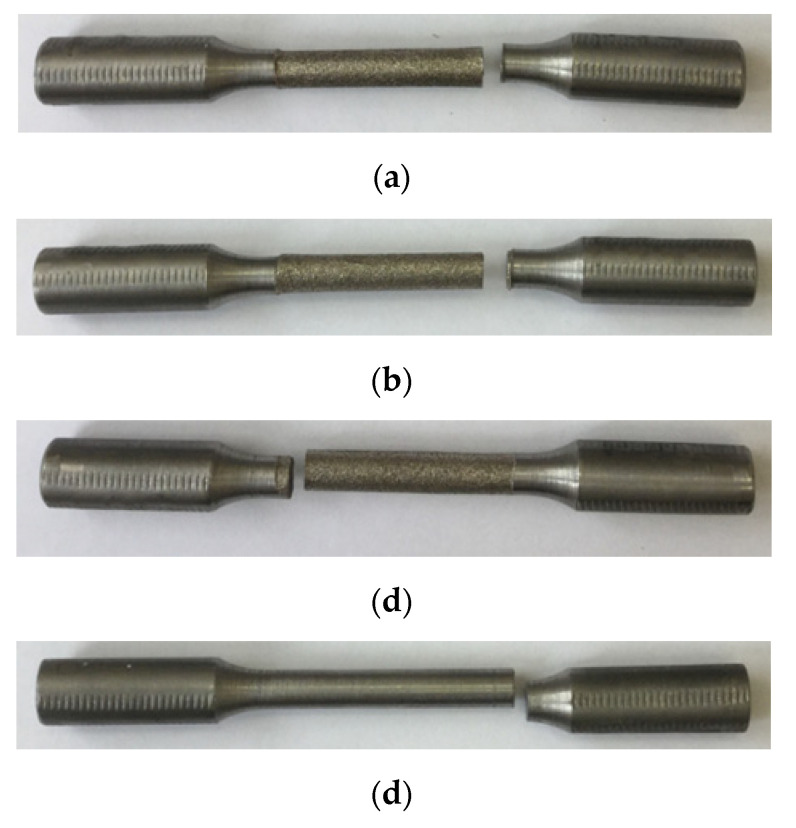
Photographs of low-cycle fatigue fractured specimens made by: (**a**) EDM (copper ET); (**b**) EDM (composite ET); (**c**) EDM (graphite ET); (**d**) turning.

**Table 1 materials-15-01566-t001:** The main directions of research of the process of EDM.

Topic of Study	Sources	Key Positions
The morphology and roughness of the surfaces after EDM	[1,2,3,4,5,7,8,12,13,14,15,16,17,18,19,20,21,22,23,24,25,26,27]	The melting of the treated surface is accompanied by a change in its structure, the grain is refined, and zones of plastic deformation appear. Surface roughness after EDM is characterized by a set of mutually intersecting single holes. The size of the individual wells depends on the charge energy.
The mechanical properties of the surfaces after EDM	[6,11,13,15,17,28,29,30,31]	As a result of EDM, there is a difference in the mechanical properties of the surface layer and the base material. Different surface conditions can affect the fatigue characteristics of the material. Residual stresses of a tensile nature are formed in the treated surface.
The chemical composition of the surfaces after EDM	[4,5,7,8,19,21,22,24,26,32,33]	During the EDM process, there is a change in the elemental composition of the surface layer. TE material has the greatest influence on the change in the composition of the surface layer. By dielectric flows, molten particles of ET material enter the melting zone of the workpiece material and mix with it.
The white layer formed on the surfaces after EDM.	[5,6,8,9,13,15,19,21,22,23,25,31,32,33]	The white layer has a fine-grained structure with high chemical resistance. The white layer after EDM has a rough surface and contains many voids, pores, and microcracks. It can radically differ from the base material not only in properties, but also in chemical composition.

**Table 2 materials-15-01566-t002:** EDM modes.

Mode	I, A	Ton, μs	U, B
Finishing	2	40	50
Rough	8	150	100

**Table 3 materials-15-01566-t003:** Material parameters.

Parametr	Value
Young’s modulus	2 × 105 MPa
Poisson’s ratio	0.28

**Table 4 materials-15-01566-t004:** OH measurement modes.

Parametr	Value
Method of measurement	modified “χ-method”
Collimator *∅*	5 mm
Directions φ to the point of analysis (Figure 6)	0° and −90°
X-ray tube anode	Cr
Vanadium filters	Not
Diffraction line (hkl)	(220)
Diffraction angle 2θ	156.7°
The penetration depth of X-ray radiation at χ = 0°	6.3 μm
Exposure time in one position of the goniometer	20 s
Tilt angles χ	in the range [−30°; 30°], symmetrical in absolute values in both directions, where positive tilt angles χ in the range [0°; 40°] and negative tilt angles −χ in the range [−40°; 0°]
Number of tilt angles ±χ	13, where N+χ = N−χ= 7 (including χ = 0° and assuming that the measurement at the position χ = 0° is carried out once)
X-ray beam oscillation (oscillation)	3°

**Table 5 materials-15-01566-t005:** Parameters of mathematical processing of OH measurement results.

Parametr	Value
Peak calculation	Peak Fit Method
Peak level used for calculation	75
Subtracting background radiation values	Linear
Setting 2θ angles	Calibrated
Calculation of principal stresses	Three-way method 0°, −90°
Stress tensor	Three-way method 0°, −90°

**Table 6 materials-15-01566-t006:** Chemical composition of samples before and after EDM.

Mode	Electrode	Fe	Mn	Si	Cu
Original		The foundation	0.6%	0.2%	-
Finishing	Cu	The foundation	0.4%	0.2%	0.5%
	C	The foundation	0.3%	-	-
	Cu-C	The foundation	0.2%	-	-
Rough	Cu	The foundation	0.6%	0.1%	2.8%
	C	The foundation	0.4%	-	-
	Cu-C	The foundation	0.3%	-	0.9%

**Table 7 materials-15-01566-t007:** Surface roughness after EDM.

Mode	Electrode Tool	Roughness Parameters, μm		
		Ra	Rmax	Sm
Finishing	Cu	4.06	25.93	275.77
	C	3.95	27.85	228.18
	Cu-C	4.16	26.23	221.76
Draft	Cu	6.99	44.69	321.09
	C	6.77	34.51	307.29
	Cu-C	6.84	42.46	279.92

**Table 8 materials-15-01566-t008:** Tensile mechanical properties of steel 15 depending on the processing method.

Processing Method	E, GPa	$\sigma_{B},\;\mathrm{MPa}$	$\sigma_{0.2},\;\mathrm{MPa}$
Turning	233	742	369
EDM (Cu)	214	716	341
EDM (C)	210	725	345
EDM (Cu-C)	247	723	355

**Table 9 materials-15-01566-t009:** Study of the durability of specimens from steel 15 at low-cycle fatigue.

Processing Method	Durability, Cycles
Turning	27,600
EDM (copper ET)	18,938
EDM (graphite ET)	21,673
EDM (composite ET)	19,044
EDM (medium)	19,885

## Data Availability

Not applicable.

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
