# Peer review of "Study of the Structure and Mechanical Properties after Electrical Discharge Machining with Composite Electrode Tools"

_materials, 2022, doi:10.3390/ma15041566_

Round 1

Reviewer 1 Report

The paper is devoted to increasing the efficiency of electrical discharge machining (EDM) of high-quality parts with a composite electrode-tool (copper, graphite, composite material of the copper-graphite system with a graphite content of 0.2%.). The topic is original. 

It is moderate to the reference comparison with other studies. The methodology of the paper seems suitable. 

Conclusion may be extended according to the results.

According to the journal expectations, references may be extended.

The tables and figures  seem appropriate.

Author Response

Authors sincerely thank the reviewer for the valuable comments and suggestions that helped to improve the quality of the paper. 

Reviewer 2 Report

In the presented work, it is always necessary to transparently express novelty and benefit. The composition of graphite and copper for the tool electrode material is commonly used in practice. However, various comparative results will certainly be beneficial and interesting for readers.

Lines 69-70: „As a rule, there is no clear difference between the zones, and in most cases they overlap each other [5,6].“ So why do we differentiate zones? The condition of the surface layer mainly affects the energy of the discharge. Its own warming and the generation of compressive stresses during EDM contribute to changes both on the surface and below the surface of the machined surface. If the machine is set up incorrectly and the wrong EDM procedure is selected, a thin hardened layer may form under the surface.

The repetition of the numbering of the chapter and subchapters (2.2) must be carefully corrected and also the legend to Fig. 18 (a, b, c, not a, b, b).

Author Response

(The authors gave the same response as above.)

Reviewer 3 Report

The authors have tried to increase the efficiency of electrical discharge machining (EDM) of high-quality parts with a composite electrode tool. The subject of research is the chemical composition of the surface layer of the processed product, microhardness, the parameter of the roughness of the treated surface, residual stresses, mechanical properties under tension, and durability with low-cycle fatigue of steel. However, there are areas of major concern.

  1. The Introduction part is too staggered. It is very difficult to comprehend as it keeps shifting focus from EDM to mechanism to surface morphology. The introduction part needs rewriting.
  2. The authors need to provide a conclusion of various sections of Table 1 where they have mentioned the literature review.
  3. The authors need to mention the composition of the composite electrode. They also need to explain how they developed this composite electrode.
  4. Figure 5 and its title are clearly mismatched. Please recheck figure 5 and its title.
  5. Please explain why and how the authors have chosen the EDM modes, variables, and their values.
  6. Section 2.2 appears twice in the manuscript. Please recheck.
  7. The section "Determination of the elemental composition of the treated surface" has no data which collaborates the author's statement.
  8. Sections 2.2 to 2.6 should be merged under a single section.
  9. Figure 9 requires proper x and y-axis.
  10. With reference to Figures 10 and 11, the authors have stated that "there is no fundamental difference in the number of microcracks". At different magnifications and field sizes, how can authors conclude that?
  11. The authors need to explain the reasons as to why the white layer thickness varied during roughing and finishing mode of processing.
  12. Section 2.4 "Microhardness of the surface layer" needs complete rewriting as it only shows how the graphs are changing. It has to also contain the scientific reasons for the changes in the behavior of the graphs.
  13. Please add the x and y-axis to figure 18.
  14. In section 3.5 " Surface roughness", please explain the reasons for behavior trends obtained in figure 18.
  15. Please add explanations regarding observations made for figure 21.
  16. The conclusions are incomplete and need complete rewriting where it highlights the significant findings of this work. 

Author Response

(The authors gave the same response as above.)

Round 2

Reviewer 3 Report

The authors have addressed most of the points but still some clarifications are needed.

  1. The introduction as mentioned earlier is still very staggered. The first 3 paragraphs of page 2 are all explaining the EDM phenomenon. But the 1st paragraph is of 2 lines, 2nd and 3rd of 5 lines each. They should be merged. Similarly on page 3, immediately below figure 2, the two paragraphs should be merged.
  2. The authors should remove the list of universities because there are a lot of other universities and technical institutes also working on EDM since 2005. That list is not needed in a technical paper at all.
  3. The response to my previous comment 3 should be added in the revised manuscript appropriately.
  4. The authors have replied that the "copper powder was mixed with a preparation of dry colloidal graphite in a ratio of 99.8/0.2." Generally, the reinforcement has to be substantial. Here the authors need to explain why such a small amount of graphite was used and what advantage was achieved by adding such marginal quantity. 
  5. The response to previous comment 11 has to be added in the revised manuscript.
  6. There are 37 references in the manuscript but not a single reference has been cited in the text. Please mention the reference in the text appropriately.

Author Response

Dear Reviewer,

I am grateful for the helpful and interesting comments by you. The comments have been addressed in the following way (Changes are highlighted in RED color in the manuscript.)
